# Comparison of the Associations between Self-Reported Sleep Quality and Sleep Duration Concerning the Risk of Depression: A Nationwide Population-Based Study in Indonesia

**DOI:** 10.3390/ijerph192114273

**Published:** 2022-11-01

**Authors:** Vivi Leona Amelia, Hsiu-Ju Jen, Tso-Ying Lee, Li-Fang Chang, Min-Huey Chung

**Affiliations:** 1School of Nursing, College of Nursing, Taipei Medical University, Taipei City 110, Taiwan; 2Department of Nursing, Faculty of Health Science, Universitas Muhammadiyah Purwokerto, Purwokerto 53182, Indonesia; 3Department of Nursing, Shuang Ho Hospital, Taipei Medical University, New Taipei City 235, Taiwan; 4Nursing Research Center, Nursing Department, Taipei Medical University Hospital, Taipei City 110, Taiwan; 5Department of Emergency and Critical Care Medicine, Taipei Medical University Hospital, Taipei City 110, Taiwan; 6Graduate Institute of Medical Sciences, National Defense Medical Center, Taipei City 114, Taiwan

**Keywords:** depression, sleep duration, sleep quality, wealth index, chronic illness, physical activity, urban, adults, Indonesia

## Abstract

There is substantial evidence that a lack of sleep quality and duration can increase the risk of depression in adults. Still, few studies have compared sleep quality and duration to the risk of depression in Indonesia. Therefore, this study aimed to compare the prevalence and risk of depression associated with both sleep quality and duration and identified those factors associated with sleep quality with sleep duration. This study was a cross-sectional study, and the data were obtained from the 2014 Indonesian Family Life Survey, with a total sample comprised of 19,675 respondents aged older than 15 years old. A self-reported questionnaire was used to assess sleep quality and duration. Depression was assessed using the Center for Epidemiologic Studies Depression (CESD-10) questionnaire. Logistic regression was used to examine the risk of depression, and multinomial logistic regression was used to examine the risk of poor sleep quality with consideration to sleep duration. The prevalence of depression was the highest in the poor sleep quality and long sleep duration groups (48.5%). After all variables associated with depression were adjusted, poor sleep quality was identified as a factor leading to a higher risk of depression (OR = 4.2; 95% CI: 3.7–4.6; *p* < 0.001) than long sleep duration (OR = 1.4; 95% CI: 1.2–1.6; *p* < 0.001). Furthermore, the interaction between poor sleep quality and long sleep duration gave the highest risk of depression (OR = 4.4; 95% CI: 3.6–5.3); *p* < 0.001). Multinomial logistic regression revealed that the factors leading to a significant increase in the risk of poor sleep quality, with consideration to sleep duration, in the population were age, gender, marital status, education, wealth index, physical activity, chronic illness, season, and urban area (*p* < 0.05). Sleep quality was found to be associated with a higher risk of depression than sleep duration. The findings of this study may be beneficial to healthcare professionals who develop health promotion strategies for reducing the incidence of depression in communities.

## 1. Introduction

Depression is a worldwide public health problem; it contributes considerably to the global illness burden and affects people from all communities. Depression can severely impair a person’s ability to manage their daily activities and can even lead to suicide [1]. The prevalence of depression in Indonesians aged 15 and older is rising annually; it increased from 6% in 2013 to 9.8% in 2018, which is greater than the global average of 4.4 [2].

The etiology of depression varies and is only partially understood, and is influenced by numerous factors. One study revealed that the occurrence of depression is related to sleep quality and sleep duration; lack of sleep can disrupt an individual’s internal biological clock in the suprachiasmatic nucleus of the hypothalamus and can thereby cause depression [3]. Sleep quality plays a key role in understanding how and why depression occurs, and sleep duration and sleep quality have been reported to be related [4,5]. Poor sleep quality can disrupt executive function systems, which directly affects the ability to manage emotions [6]. Difficulty with emotion regulation accounts for many of the contemporaneous and prospective associations between sleep quality and depressive symptoms [5]. The average sleep duration has decreased by 1–2 h—most notably in teenagers, young adults, and older adults—over the past decade [7]. Several studies have discussed the association between short [8,9] and long [10,11] sleep duration and depression.

The dual process model of the incidence of depression considered from the perspective of sleep–wake cycle regulation includes sleep quality and duration [12]. There are three main factors included in this model (sociodemographic factors, behavioral and health indicators, and environmental factors), which can affect sleep and potentially lead to depression [12]. Sociodemographic factors, such as age, gender, marital status, educational level, and economic status, have been identified as key variables in explaining variation in the prevalence of depression. Depressive symptoms were reported to be more common in younger people than in older people [13]; however, older adults were reported to be more likely than younger people to experience depression [14]. In addition, depression is more common in women than in men [15], although a study found that men also tend to experience depression [14]. Another demographic factor is marital status; being married rather than single was linked to a lower risk of depression [15]. Adults who have better education levels are less likely to have depression symptoms [13]. Individuals with a lower socioeconomic status are more likely to experience depressive symptoms, and the risk of depression reportedly decreased considerably with each level of increase in the economy index [16].

Behavioral and health indicators can also affect sleep; for example, individuals who do not engage in sufficient physical activity are more likely to experience disturbed sleep [17] and eventually become depressed [18]. Moreover, chronic illness and body mass index (BMI) are health indicators that have a correlation with depression. Studies have found that the incidence of depression is higher in people with obesity [19] and that several characteristics of chronic illnesses, such as severity and duration, are linked to sleep disruption [20] and depressive outcomes [21].

Environmental factors can lead to depression, although these factors are changeable [22]. Changes in season can affect mood; in rainy seasons, people are more likely to feel sad, which can lead to depression [23]. Another influential environmental factor is light pollution, a major environmental hazard that places ecosystems under stress and can cause depression [24]. Urbanization affects mental health by increasing stress and factors, such as overcrowding and environmental pollution, levels of violence, and a lack of social support [25]. More densely constructed environments and increases in the intensity of human interactions and their subsequent adaptive behaviors have been referred to as urban indices [26].

Numerous studies have investigated the variables linked to depression, including sleep quality [4,5,27] and sleep duration [8,9,11,28]. However, few studies have compared the association between sleep quality and duration and the risk of depression. Studies have discussed the relationship between the two variables as well as the variables’ associations with depression [29,30], though these studies have mostly focused on sleep duration and quality in either older adults [30] or young adults [29]. Several studies were conducted in Indonesia that analyzed the associated factors of the risk of depression in the community but did not include the sleep variables [31,32,33], and the population consisted only of senior adults [31], young adults [32], and adults [33]. Thus, this study aims to compare the associations between the prevalence and risk of depression and sleep quality and duration and to identify the risk factors for poor sleep quality and duration in all age groups (from teenagers to older adults). Furthermore, using a dual process model approach, this study attempted to analyze factors associated with sleep quality while taking sleep duration into consideration.

## 2. Materials and Methods

### 2.1. Data Source and Participants

This was a cross-sectional study from secondary data of the fifth wave of the Indonesian Family Life Survey (IFLS 5) in 2014. This dataset comprised anonymous data that are available to researchers who comply with research and development (RAND) corporation guidelines. The IFLS survey used stratified the sampling for rural or urban locations in 13 of the 27 provinces. Each province had enumeration areas (EAs) drawn at random from a nationally representative sample. In 1993, 321 EAs were chosen at random from 13 provinces to participate in the IFLS. A total of 20 households were recruited for each urban EA and 30 households from the rural EA. This dataset included about 83% of Indonesians residing within 321 EAs [34].

The original sample comprised 34,635 respondents aged older than 15 years old. Responses from participants, who provided age, gender, marital status, educational level, wealth index, Body Mass Index (BMI), seasonal influence, light pollution, area of residence, physical activity level, chronic illness, sleep duration, sleep quality, and depression data, were included for further analysis. Respondents with 20% missing data were excluded [35], with the resulting 19,657 respondents analyzed. The characteristics of the IFLS 5 respondents (excluded due to missing values for several variables) were compared to the characteristics of the respondents included in the analysis. It was revealed that the differences between both groups were not significant (*p* > 0.05).

The Institutional Review Board (IRB) of the RAND Corporation and Universitas Gadjah Mada in Indonesia assessed and approved the surveys and methodology for the IFLS, and respondents over the age of 18 signed individual informed consent, and for those under 15–18 years old respondents, this was given by their guardian [34]. This study also received IRB approval from the Joint Institutional Review Board of Taipei Medical University (N202107082).

### 2.2. Instruments

#### 2.2.1. Self-Reported Sleep Quality

Self-reported sleep quality was measured through responses to the item “my sleep quality was…,” with the five possible answers being very poor, poor, fair, good, and very good. The responses were then placed into three categories indicating poor (very poor and poor), intermediate (fair), and good (very good and good) sleep quality [30].

#### 2.2.2. Sleep Duration

Sleep duration was measured with the items “What time did you wake up yesterday?” and “What time did you go to sleep yesterday?” Individuals with a sleep duration within the National Sleep Foundation (NSF) recommended range (8 to 10 h for teenagers (14–17 years), 7 to 9 h for young adults (18–24 years), and adults (26–64 years), and 7 to 8 h for older adults (≥65 years) were considered to have normal sleep duration [36]. Individuals with a lower or higher sleep duration than the NSF recommended range were considered to have a shorter than normal and longer than the normal sleep duration, respectively.

#### 2.2.3. Depression

Depression symptom was assessed through the 10 items of the Center for Epidemiological Studies Depression (CESD-10) questionnaire. This questionnaire is commonly used to assess depressive symptoms in adults [37]. Each item has four possible answers that are rated on a Likert scale (0–3), with total scores ranging from 0 to 30. A total score <10 indicates no depression, and a score ≥10 indicates depression. With a Cronbach’s α of 0.86 and an interclass correlation of 0.85, the CESD-10 has demonstrated high internal consistency and test–retest reliability [34]. For the Indonesian version, the Cronbach’s α coefficient is 0.9 [38]. The Cronbach’s α in this study was 0.72.

#### 2.2.4. Sociodemographic, Lifestyle, Health Indicators, and Environmental Exposure

Age, gender, marital status, education, and wealth index were the five sociodemographic characteristics considered in this study. Education is categorized by low (<12 years attainment) and high (≥12 years attainment). The wealth index is a household expenditure aggregate that includes information gathered at the individual and household levels, as well as multiple indicators of economic and noneconomic well-being, such as consumption, income, and assets [34]. In this study, the wealth index was coded from 1 to 5, which were assigned to the poorest and richest quintiles, respectively [34].

For the lifestyle and health indicators, levels of physical activity were assessed through items from a modified short-form version of the International Physical Activity Questionnaire. The items were used to assess the type and duration of physical activity involved in all aspects of life, such as that involved in work, homelife, and exercise, and were subsequently classified as indicating low, moderate, or high levels of exercise [34]. Chronic illness was assessed through self-rated health questionnaire items; respondents answered “yes” or “no” to having various diseases; if they answered “yes” for one or more diseases, they were considered to have a chronic illness, including diabetes, tuberculosis, asthma, lung disease, heart disease, liver, stroke, cancer, prostate illness, and kidney disease [34]. BMI categorizations were based on The Ministry of Health of Indonesia (underweight (<17.0 kg/m^2^), thin (17.0–18.4 kg/m^2^), normal (18.5–25.0 kg/m^2^), overweight (25.1–27.0 kg/m^2^), and obese (>27 kg/m^2^)), were used [39].

Three variables were used to assess environmental exposure. The first variable was seasonal factors, which were evaluated according to the province where the respondents lived, the month and year when the data were collected; these data were further divided into two categories: dry and rainy seasons. The second variable was light pollution, which was retrieved and tagged using the Quantum Geographical Information System (QGIS) open-source geographic information application based on the Visible Infrared Imaging Radiometer Suite (VIIRS) 2014 data observed for the sky radiance at night and the latitude and longitude of participants’ provinces. [40]. The measurement of nighttime radiance (1 radiance unit = 10^−9^ W/cm^2^/sr) was used to assess light pollution. Levels of light pollution were categorized using a light pollution map from: https://www.lightpollutionmap.info/, eight grades of exposure were identified for the included counties: 0.00–0.15 units, 0.15–0.25 units, 0.25–0.50 units, 0.50–1.50 units, 1.50–10.00 units, 10.00–50.00 units, 50.00–75.00 units, and ≥75.00 units [40]. The third was urban residence, which takes into account population density, the proportion of residents involved in agricultural labor, but instead access to amenities such as healthcare and education [34].

### 2.3. Statistical Analysis

Descriptive analyses of participant characteristics were presented in the form of numbers, percentages, and graphs. A bivariate analysis to identify associations between each variable and depression was conducted using chi-square tests. Multivariate logistic regression was used to analyze the risk of depression, and the adjusted model included the following variables: age, gender, educational level, marital status, wealth index, physical activity level, chronic illness, BMI, season, light pollution, and area of residence. Multinomial logistic regression analysis was applied to predict the risk of poor sleep quality with consideration of sleep duration. The percentage of the odds ratio (OR) was reported using a 95% confidence interval (CI). Statistical Product and Service Solutions (SPSS), version 24 (SPSS Inc., Chicago, IL, USA), Windows, was used for all analyses.

## 3. Results

Overall, of the 19,657 respondents studied, 4429 (22.5%) had depression. Table 1 shows the sociodemographic and other variables of depressed and nondepressed respondents. In terms of sociodemographic factors, the majority of respondents (72.5%) were aged 26 to 64, and females comprised 53.8%. The proportion of respondents who were married or had previously been married was 80.5%, and the majority had low education (52.2%). The proportion of respondents who had a low wealth index was 22.6%. Furthermore, in the lifestyle and health indicator factors, the proportion of respondents with low levels of physical activity was 53.4%. The majority of respondents (69%) were free of chronic illness, and the respondents who had a BMI between 18.5 and 25 kg/m^2^, comprised 55.7%. For environmental exposure, the rainy season had the highest proportion (72.6%). The majority of respondents (35.7%) lived in a light pollution area, with pollution levels ranging from 1.5 to 10 units. Moreover, respondents living in urban areas made up 60.6%. The majority of the respondents (56.2%) reported intermediate sleep quality and normal sleep duration (57.4%). Additionally, age, marital status, wealth index, physical activity, chronic illness, BMI, season, light pollution, area of residence, sleep quality, and sleep duration were found to be significantly related to depression, but not gender and education.

After adjusting for all factors, Table 2 shows that sleep duration and sleep quality were significantly associated with a higher risk of depression. Compared to respondents with good sleep quality, respondents with poor (OR = 4.2; 95% CI: 3.8–4.7; *p* < 0.001) and intermediate sleep quality (OR = 1.4; 95% CI: 1.3–1.5; *p* < 0.001) had a higher risk of depression. Compared to normal sleep duration, respondents with long (OR = 1.4; 95% CI: 1.2–1.6; *p* < 0.001) and short sleep duration (OR = 1.1; 95% CI: 1.0–1.1; *p* = 0.029) had a higher risk of depression. Moreover, the interaction of sleep quality and sleep duration showed that the interaction of poor sleep quality and long sleep duration had the highest risk of depression (OR = 4.4; 95% CI: 3.6–5.3; *p* < 0.001) compared to the interaction of good sleep quality and normal sleep duration.

The percentages for the prevalence of depression for each sleep quality group with consideration for sleep duration are presented in Figure 1. The prevalence of depression was highest in the poor sleep quality and long sleep duration group (48.5%). All sleep duration groups with poor sleep quality had higher incidences of depression than the other sleep quality groups.

The risk factors for poor and intermediate sleep quality in the short- and long-sleep duration groups are presented in Table 3. The variable that significantly increased the risk of poor sleep quality in the short sleep duration group was the age variable. Education, season factors, and living in an urban area significantly increased the risk of poor and intermediate sleep quality in all groups of all sleep durations. The variables that significantly increased the risk of poor sleep quality in all of the groups of sleep duration were gender, marital status, wealth index, physical activity, and chronic illness.

## 4. Discussion

In this large, nationally representative population sample, individuals with poor sleep quality and long sleep duration had the highest prevalence of depression (48.5%; Figure 1). This current study revealed that sleep quality was found to be a higher risk of depression than sleep duration. On the other hand, respondents with poor sleep quality and long sleep duration had the highest risk of depression. Furthermore, the dual process model approach was used to examine the factors that significantly increased the risk of poor sleep quality when sleep duration was considered: gender, marital status, education, wealth index, physical activity, chronic illness, season, and urban area.

This study showed that individuals with poor sleep quality and long sleep duration had a higher risk of depression (OR = 4.4; 95% CI: 3.6–5.3; *p* < 0.001; Table 2). Moreover, poor sleep quality had the highest risk of increasing the incidence of depression (OR = 4.2; 95% CI: 3.8–4.7; *p* < 0.001) and intermediate sleep quality had an OR of 1.4 (95% CI: 1.2–1.6; *p* < 0.001; Table 2). This finding is consistent with that of a previous study that indicated that individuals with poor sleep quality develop mood disorders, which were recurrent over the study’s 1-year follow-up period [41]. Sleep quality has a strong association with depression; poor sleep quality affects emotion regulation and contributes to depression [5]. Poor sleep quality was also associated with greater difficulty in disengaging attention from negative stimuli, which can exacerbate depressive symptoms [6]. Another study revealed that sleep quality, not sleep duration, significantly increases the levels of depression; this was attributed to the workload effect, which prevents individuals from falling asleep and from feeling refreshed after waking up [42].

Long sleep duration was associated with a higher risk of depression (OR = 1.4; 95% CI = 1.2–1.6; *p* = 0.029) than short sleep duration (OR = 1.1; 95% CI: 1.2–1.6; *p* < 0.001; Table 2). This finding was similar to that of a study in which a large community survey revealed long sleep duration to be significantly associated with depression [11]. Long sleep duration has also been reported to be associated with depression in teenagers [10], as well as in adults [29]. According to another study, a U-shaped association was reported between sleep duration and depressive events, which suggests that individuals with both short and long sleep durations have a higher incidence of depression [29]. Long sleep durations increase inflammation levels, which are reportedly higher in individuals with depression, indicating that low-grade chronic inflammation may be a key physiological route linking sleep and depressive symptoms [43].

Furthermore, the dual process model approach was used to investigate those factors that increase the risk of poor sleep quality when considering sleep duration, such as age, gender, marital status, education, wealth index, physical activity, chronic illness, season, and urban area (Table 3). Having an age of more than 26 years increased the risk of poor sleep quality, this finding is similar to a previous study that found that older people are more likely to have poorer sleep quality than younger people [17]. Individuals at a productive age have many things that they have to do that lead them to have sleep problems, and older adults experience a reduced functional body system [44]. Female gender is one of the factors of poor sleep quality; this finding is consistent with a previous study which states that females have greater rates of comorbid insomnia than males [17]. According to one study, premenstrual dysmorphic disorder, the process that occurs one week before the menstrual cycle, usually occurs during the luteal phase; this process can cause sleep disturbance in women [45]. The other factor is marital status: this finding was similar to a previous study that said being single/unmarried/divorced was more likely to have sleep disturbance [17], and intimate partners or spouses are a primary source of social control when it comes to bedtime or medication administration [46]. Close relationships affect one’s physical health and well-being [46].

Having a low education level and being unwealthy are more likely to affect poor sleep quality; this finding is similar to that of a previous study in which it was stated that good education and a high income have more beneficial effects on an individual’s health status than those who are less educated, unemployed, and low income [17]. A higher level of education can give the individual more possibilities for getting a job and an income, which can make them secure and has the effect of having more and better sleep [45]. Moreover, the wealth index is one factor for increasing poor sleep quality when considering sleep duration; this finding is in line with that of another study, which indicated that individuals with a lower socioeconomic status reported lower sleep quality and improvements in socioeconomic status and neighborhood quality resulted in fewer sleep problems [17].

Low levels of physical activity were associated with a higher risk of poor sleep quality for both short and long sleep durations; this finding is consistent with that of another study that demonstrated that the level of physical activity was a significant factor affecting difficulty in falling asleep [47]. Rapid eye movement (REM) sleep is considered to be an indicator of sleep quality, and low levels of exercise are strongly associated with a reduction in REM sleep [48]. Increasing physical activity was positively linked with sleep restoration, and vigorous physical activity levels were reported to be a stronger predictor of high-quality sleep than moderate physical activity levels were [49]. Improving physical activity through both the performance of exercises and a willingness to engage in daily physical activities may improve sleep quality. Furthermore, chronic diseases have been linked to sleep; the symptoms of the disease may lead to discomfort, which affects sleep quality [50]. Patients with chronic medical conditions frequently receive fewer hours of sleep and less restorative sleep than healthy individuals [20]. Individuals with COPD often have poor sleep quality due to desaturation episodes occurring during REM sleep; these episodes are caused by atonia in the skeletal muscles and the accessory muscles for respiration [20].

The rainy season increases the risk of poor sleep quality. This finding is in line with a previous study which stated that it was the main factor contributing to the seasonality of sleep duration [51]. In the rainy season, a person will have a longer sleep duration, but in the summer, they will have a shorter sleep duration [51]. Moreover, residence in an urban area had a significant association with an increased risk of poor sleep quality, with consideration to sleep duration (Table 3). This finding is consistent with that of another study, which revealed that noise, traffic, and urban density, are all linked to poor sleep quality, which can lead to lower sleep efficiency and more awakenings [52]. Residence in an urban area can affect healthy sleep patterns; characteristics of urban neighborhoods, such as walkability, number of green spaces, and urban density were linked to sleep duration, daytime drowsiness, sleep problems, and sleep quality [53].

## 5. Limitations

This study used a large sample from a national survey which was representative of the Indonesian population. Furthermore, the study included several age categories, including teenagers, young adults, adults, and older adults. However, this study had several limitations. First, the study method was cross-sectional because the variables included in the study were obtained only from wave five of the IFLS, consequently making it difficult to elucidate the causal relationship between the sleep-related variables and depression. In future studies, the inclusion of an appropriate dataset would enable a longitudinal study method to investigate the long-term changes in sleep quality and sleep duration. Second, the sleep quality and depressive symptom data obtained in this study were self-reported, although self-reported sleep quality was employed in another study [30], and the CESD-10 items have been frequently used to assess depression and depressive symptoms [37]. Third, sleep duration data were measurements taken “at the time” and, therefore, did not reflect weekday and weekend sleep durations. However, another study also adopted similar time measurements for sleep duration [29]. The recommendations for measuring sleep duration do not specify the need for separate weekday and weekend measurements [36].

## 6. Conclusions

This study provided evidence that sleep quality, rather than sleep duration, leads to a higher risk of depression. Depression was more common in the sleep duration groups with poor sleep quality than in the other sleep quality groups. Furthermore, this study identified the risk factors (age, gender, marital status, education, wealth index, physical activity, chronic illness, season, and urban area) that significantly in-creased the risk of poor sleep quality, with consideration to sleep duration.

## 7. Implications

It is challenging to ascertain people with depressive symptoms from a community, even though family and community are the first lines of defense in assessing and preventing depression. From nursing, clinical, and public health perspectives, these findings may be important when screening for the risk of depression in adolescents and adults. Additionally, this study can be useful for planning health promotion strategies because sleep factors are modifiable through sleep improvement interventions, such as music relaxation [54], environmental improvements [52], and lifestyle improvements, such as physical activity [49]. The findings may have important implications for future research into sleep-based depression mechanisms. A subsample of participants would be chosen for further research in order to cross-check their responses with a more objective empirical measure based on technological resources that assess sleep quality, such as actigraphy, while also considering the influence of schoolwork and parents on the development of sleep problems in teenagers or the influence of cultural background on the development of sleep problems.

## Figures and Tables

**Figure 1 ijerph-19-14273-f001:**
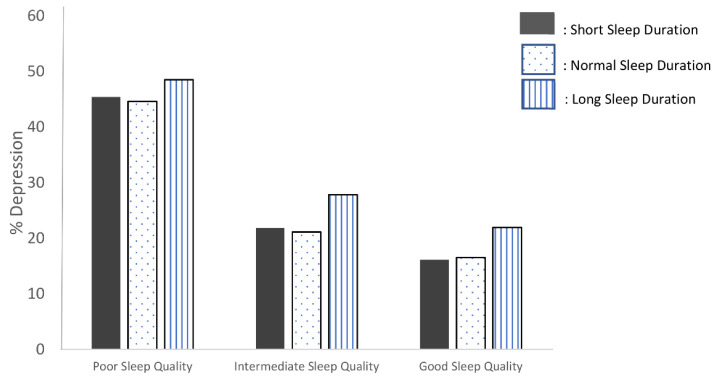
Percentage of depression prevalence associated with sleep duration and quality.

**Table 1 ijerph-19-14273-t001:** Variables for participants with and without depressive symptoms (*n* = 19,657).

	Total	Depressed (*n* = 4429)	Non-Depressed (*n* = 15,228)	*p* Value
	***n* (%)**	***n* (%)**	***n* (%)**	
**Socio-demographics**				
Age (years)				<0.001
15–17	1531 (7.8)	436 (9.8)	1095 (7.2)	
18–25	3581 (18.2)	1053 (23.8)	2528 (16.6)	
26–64	14,258 (72.5)	2907 (65.6)	11,351 (74.5)	
≥65	287 (1.5)	33 (0.7)	254 (1.7)	
Gender				0.100
Male	9090 (46.2)	2000 (45.2)	7090 (46.6)	
Female	10,567 (53.8)	2429 (54.8)	8138 (53.4)	
Marital status				<0.001
Single/never married	3828 (19.5)	1128 (25.5)	2700 (17.7)	
Married/ever married	15,829 (80.5)	3301 (74.5)	12,528 (82.3)	
Education				0.076
Low (<12 years attainment)	10,255 (52.2)	2535 (53.1)	7902 (51.9)	
High (≥12 years attainment)	9402 (47.8)	2076 (46.9)	7326 (48.1)	
Wealth index				<0.001
Q1: poorest	4433 (22.6)	1093 (24.7)	3340 (21.9)	
Q2	4271 (21.7)	948 (21.4)	3323 (21.8)	
Q3	4053 (20.6)	927 (20.9)	3126 (20.5)	
Q4	3831 (19.5)	804 (18.2)	3027 (19.9)	
Q5: least poor	3069 (15.6)	657 (14.8)	2412 (15.8)	
**Lifestyle and health indicators**				
Physical activity level				<0.001
Low	10,488 (53.4)	2139 (48.3)	8349 (54.8)	
Moderate	6518 (33.2)	1560 (35.2)	4958 (36.6)	
High	2651 (13.5)	730 (16.5)	1921 (12.6)	
Chronic illness				<0.001
No	13,559 (69.0)	2828 (63.9)	10,731 (70.5)	
Yes	6098 (31.0)	1601 (36.1)	4497 (36.1)	
BMI				<0.001
≤18.4	2150 (10.9)	542 (12.2)	1608 (10.6)	
18.5–25.0	10,949 (55.7)	2536 (57.3)	8413 (55.2)	
25.1–27.0	2379 (12.1)	489 (11)	1890 (12.4)	
>27	4179 (21.3)	862 (19.5)	3317 (21.8)	
**Environmental exposure**				
Season				0.009
Dry	5378 (27.4)	1143 (25.8)	4235 (27.8)	
Rainy	14,279 (72.6)	3286 (74.2)	10,993 (72.2)	
Light pollution level				<0.001
0.00–0.15	914 (4.6)	203 (4.6)	711 (4.7)	
0.15–0.25	251 (1.3)	46 (1)	205 (1.3)	
0.25–0.50	1417 (7.2)	256 (5.8)	1161 (7.6)	
0.50–1.50	6493 (33)	1562 (35.3)	4931 (32.4)	
1.50–10.00	7027 (35.7)	1594 (36)	5433 (35.7)	
10.00–50.00	3241 (16.5)	691 (15.6)	2550 (16.7)	
50.00–75.00	131 (0.7)	30 (0.7)	101 (0.7)	
>75	183 (0.9)	47 (1.1)	136 (0.9)	
Area of residence				0.032
Urban	11,910 (60.6)	2745 (62)	9165 (60.2)	
Rural	7747 (39.4)	1684 (38)	6063 (39.8)	
**Sleep quality**				<0.001
Poor	2129 (10.8)	963 (7.7)	1166 (21.7)	
Intermediate	11,053 (56.2)	2381 (56.9)	8672 (53.8)	
Good	6475 (32.9)	1085 (35.4)	5390 (24.5)	
**Sleep duration**				<0.001
Short	7019 (35.7)	1603 (36.2)	5416 (35.6)	
Normal	11,275 (57.4)	2441 (55.1)	8834 (58)	
Long	1363 (6.9)	385 (8.7)	978 (6.4)	

**Table 2 ijerph-19-14273-t002:** Association among sleep duration, sleep quality, and depression.

	Model 1 OR (95%CI)	*p*	Model 2 OR (95%CI)	*p*	Model 3 OR (95%CI)	*p*	Model 4 OR (95%CI)	*p*
Sleep quality								
Poor	4.1 (3.7–4.6)	<0.001	-	-	-	-	4.2 (3.8–4.7)	
Intermediate	1.3 (1.2–1.5)	<0.001	-	-	-	-	1.4 (1.3–1.5)	
Good	1		-	-	-	-	1	
Sleep duration								
Short	-	-	1.1 (1.0–1.2)	0.027	-	-	1.1 (1.0–1.1)	0.029
Normal	-	-	1		-	-	1	
Long	-	-	1.3 (1.2–1.6)	<0.001	-	-	1.4 (1.2–1.6)	<0.001
Sleep quality × sleep duration								
Good SQ × Normal SD	-	-	-	-	1		1	
Good SQ × Short SD	-	-	-	-	0.9 (0.9–1.1)	0.904	0.9(0.9–1.1)	0.904
Good SQ × Long SD	-	-	-	-	1.4 (1.2–1.6)	0.001	1.3(1.1–1.6)	0.001
Intermediate SQ × Short SD	-	-	-	-	0.7 (0.6–0.8)	<0.001	0.6(0.6–0.8)	<0.001
Intermediate SQ × Long SD	-	-	-	-	1.1 (0.8–1.3)	0.866	1.0(0.9–1.1)	0.986
Poor SQ × Short SD	-	-	-	-	2.6 (1.7–3.9)	<0.001	2.5(1.6–3.8)	<0.001
Poor SQ × Long SD	-	-	-	-	4.5 (3.6–5.3)	<0.001	4.4(3.6–5.3)	<0.001

OR = Odds Ratio: Model 1, Model 2, Model 3, Model 4: adjusted with all sociodemographics, lifestyle and health indicators and environmental exposure variables.

**Table 3 ijerph-19-14273-t003:** Results of the multinomial logistic regression analysis of variables predicting sleep quality.

	Short Sleep Duration (*n* = 7019)		Long Sleep Duration (*n* = 1363)	
	Poor (*n* = 930) vs. Good Sleep Quality (*n* = 2143) OR (95%CI)	Intermediate (*n* = 3946) vs. Good Sleep Quality (*n* = 2143) OR (95%CI)	Poor (*n* = 169) vs. Good Sleep Quality (*n* = 487) OR (95%CI)	Intermediate (*n* = 707) vs. Good Sleep Quality (*n* = 487) OR (95%CI)
**Socio-demographics**				
Age (years)				
15–17	1	1	1	1
18–25	0.9 (0.5–1.8)	0.9(0.6–1.5)	0.5 (0.2–1.0)	0.9 (0.4–1.2)
26–64	**1.5 (1.0–3.1) ***	1.2(0.7–1.9)	1.2 (0.4–1.4)	1.6 (0.6–1.8)
≥65	**0.4 (0.1–0.8) ***	0.9(0.6–1.4)	1.2 (0.5–1.5)	1.4 (0.7–1.8)
Gender				
Male	1	1	1	1
Female	**1.1 (0.9–1.3) ***	0.9(0.9–1.1)	**1.8 (1.2–2.0) ***	1.1 (0.8–14)
Marital status				
Single/never married	**1.1 (1.0–1.9) ***	0.9(0.7–1.2)	**1.2 (1.1–1.6) ***	0.9 (0.6–1.3)
Married/ever married	1	1	1	1
Education				
Low (<12 years attainment)	**1.9 (1.7–2.4) ***	**1.7 (1.5–1.9) ***	**1.9 (1.4–2.1) ***	**1.7 (1.4–2.3) ***
High (≥12 years attainment)	1	1	1	1
Wealth index				
Q1: poorest	**1.1 (1.0–1.5) ***	**1.1 (1.0–1.6) ***	**1.5 (1.2–1.9) ***	1.1 (0.7–1.6)
Q2	0.9 (0.7–1.2)	0.9 (0.7–1.0)	1.3 (0.7–1.5)	1.2 (0.8–1.5)
Q3	0.9 (0.7–1.2)	0.9 (0.7–1.1)	0.8 (0.4–1.3)	0.9 (0.6–1.3)
Q4	0.6 (0.4–1.2)	0.8 (0.7–1.1)	0.7 (0.4–1.3)	0.8 (0.6–1.2)
Q5: least poor	1	1	1	1
**Lifestyle and health indicators**				
Physical activity level				
Low	**1.8 (1.4–2.3) ***	1.1 (0.9–1.3)	**1.3 (1.0–1.5) ***	1.0 (0.7–1.3)
Moderate	1.1 (0.9–1.4)	0.9 (0.8–1.1)	1.1 (0.7–1.4)	0.9 (0.6–1.3)
High	1	1	1	1
Chronic illness				
No	1	1	1	1
Yes	**1.2(1.0–1.5) ***	0.9 (0.8–1.0)	**1.1 (1.0–1.7) ***	**0.7 (0.5–0.9) ***
BMI				
≤18.4	1.0 (0.7–1.3)	1.1 (0.8–1.1)	1.3 (0.7–1.1)	1.2 (0.8–1.6)
18.5–25.0	1	1	1	1
25.1–27.0	1.1 (0.8–1.5)	0.9 (0.7–1.1)	0.7 (0.4–1.3)	1.2 (0.7–1.5)
>27	1.1 (0.9–1.3)	1.2 (1.0–1.5)	0.9 (0.5–1.4)	1.1 (0.6–1.5)
**Environmental exposure**				
Season				
Dry	1	1	1	1
Rainy	**1.2 (1.0–1.4) ***	**1.1 (1.0–1.3) ***	**1.3 (1.0–1.5) ***	**1.4 (1.1–1.7) ***
Light pollution level				
0.00–0.15	1	1	1	1
0.15–0.25	0.9 (0.4–1.1)	0.9 (0.4–1.3)	0.4 (0.1–1.2)	0.3 (0.1–1.0)
0.25–0.50	1.3 (0.6–1.6)	0.8 (0.7–1.1)	0.5 (0.3–1.1)	0.5 (0.4–1.3)
0.50–1.50	1.1 (0.5–1.2)	0.8 (0.4–1.4)	0.5 (0.4–1.2)	0.7 (0.5–1.3)
1.50–10.00	1.2 (0.6–1.5)	0.8 (0.5–1.3)	0.6 (0.5–1.1)	0.4 (0.3–1.4)
10.00–50.00	1.4 (0.6–1.7)	0.9 (0.5–1.4)	0.6 (0.3–1.2)	0.9 (0.6–1.5)
50.00–75.00	**1.6 (1.2–1.9) ***	0.9 (0.5–1.5)	0.9 (0.6–1.3)	1.1 (0.4–1.5)
>75	**2.2 (1.5–3.1) ***	**2.8 (1.1–3.4) ***	-	-
Area of residence				
Urban	**1.4 (1.0–1.7) ***	**1.2 (1.0–1.3) ***	**1.2 (1.0–1.5) ***	**1.1 (1.0–1.7) ***
Rural	1	1	1	1

* *p* < 0.05.

## Data Availability

The data used in this study were obtained from the RAND Corporation’s secondary dataset for the Fifth Wave of the Indonesian Family Life Survey (2014); more information can be found at https://www.rand.org/well-being/social-and-behavioral-policy/data/FLS/IFLS.html (accessed on 24 March 2021). The RAND Corporation must approve the use of this dataset.

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
