# Peer review of "Comparison of the Associations between Self-Reported Sleep Quality and Sleep Duration Concerning the Risk of Depression: A Nationwide Population-Based Study in Indonesia"

_ijerph, 2022, doi:10.3390/ijerph192114273_

Round 1
Reviewer 1 Report
The study has a large enough sample to be considered representative of the population to which it refers.
Likewise, I consider the logistic regression model to be adequate for the analysis of the data.
I do think it would be appropriate to indicate how the terms that may represent interactions between the predictor variables have been taken into account in the model.
On the other hand, perhaps the greatest weakness observed is the way in which the sleep quality variable is operationalised, given that it is totally subjective (indeed, you indicate this as a limitation of the study).
As a suggestion, one possibility for future studies would be to select a sub-sample of participants in order to be able to cross-check their responses with a more objective empirical measure based on technological resources that assess sleep quality.
Finally, when reading the conclusions associating long sleep duration with the presence of depression. Perhaps it should be pointed out that the type of design used does not allow us to be sure whether this longer duration of sleep is a consequence of the presence of a depressive state or an antecedent of such a depressive state.
This is one of the advantages that would be obtained using the longitudinal studies referred to.
I think the implications are well presented and it is a good contribution to the design of preventive programmes.
Author Response
Dear Editor and Reviewer,
Thank you for allowing us to revise our manuscript and for the helpful suggestions that helped us improve the article. All suggested changes have been incorporated into the manuscript, and the revised sections have been highlighted.

Reviewer 2 Report
Thank you for providing the opportunity to review this manuscript. This is an excellent study exploring factors associated with depressive symptoms among the Indonesian population. I have a few comments that might help to improve this study, as follows.
Background
1. Overall, the Introduction section looks good. However, I found a similar study that has been published using the same dataset (IFLS 5) (https://doi.org/10.1016/j.npbr.2018.04.004). The authors need to justify whether their present study is different from that published study. The author might need to cite that study also in the Introduction section and elaborate on what this present study is going to add/improve the current evidence from Indonesia.
2. Please also look at previous studies using IFLS 5 to investigate factors associated with depressive symptoms in Indonesia (https://www.rand.org/well-being/social-and-behavioral-policy/data/FLS/IFLS/papers.html). The authors need to improve their background by summarising findings from the Indonesian context and adding information on the novelty of their study.
Methods
3. In the sub-section of data source and participants, please add a figure/ flow diagram that illustrates the exclusion of participants in this study (including the initial number of IFLS participants, excluded participants, and final sample size).
4. The authors might need to address the issue of missingness since missing values can produce biased estimates due to potential selection bias. The authors need to add information on differences in sociodemographic characteristics between those who were included in this analysis (analytical sample) and those who were not. In addition, this study might benefit from using an inverse probability weighting (IPW) approach (for example, see http://sites.utexas.edu/prc/files/IPWRA.pdf) to address missing values in the dataset and potential selection bias. This probability weighting should be combined/incorporated with the sample weights available in IFLS 5 to improve the representativeness of the findings.
5. Please add information on the sampling method of IFLS 5.
6. The authors need to mention how IFLS 5 obtained informed consent from the participants, particularly those who were under 18 years old.
7. The authors need to add a variable of educational status in their analyses. A lot of studies using IFLS 5 data included this variable as an additional proxy of SES.
8. Please mention what types of chronic diseases were included in defining chronic illness.
9. Please mention at what level of geographical areas/boundaries that sessional factors or light pollution were defined (e.g., province, district, sub-district?). In addition, the authors need to justify why they used eight categories for grading light pollution.
10. As suggested above, the authors might need to consider including sample weights in their analysis, as well as using the IPW approach.
Results
11. To further explore the adjusted association between sleep quality, duration and depressive symptoms, the authors might need to add two-way interaction terms between sleep quality and duration in predicting depressive symptoms. This will provide estimates of OR of having depressive symptoms for participants with poor sleep quality and short/long sleep duration, relative to those with good sleep quality and normal sleep duration.
12. For Table 3, please add the sample size for each multinomial regression model since the analyses were stratified by sleep duration.
13. Please elaborate on whether Table 2 presents findings from separate regression models between sleep quality and sleep duration or whether both sleep variables were included in the same model.
Discussion
14. The authors have discussed the findings from their study very well. The authors could have discussed more the findings related to poor sleep quality with short duration among participants living in areas with exposure to the light pollution of 50-70.
15. In addition, findings on the associations between age, marital status, and sleep are interesting to be discussed.
Author Response

(The authors gave the same response as above.)

Round 2
Reviewer 2 Report
I have no further comments for this manuscript.